biomechanics/behaviour/evolution

domestication, wild boar, phenotypic plasticity, locomotion, geometric morphometrics, experimentation

# The mark of captivity: plastic responses in the ankle bone of a wild ungulate (*Sus scrofa*)

Hugo Harbers[1], Dimitri Neaux[1], Katia Ortiz[2], Barbara Blanc[2], Flavie Laurens[3], Isabelle Baly[3], Cécile Callou[3], Renate Schafberg[4], Ashleigh Haruda[4], François Lecompte[5], François Casabianca[6], Jacqueline Studer[7], Sabrina Renaud[8], Raphael Cornette[9], Yann Locatelli[2], Jean-Denis Vigne[1], Anthony Herrel[10] and Thomas Cucchi[1]

[1]UMR 7209, Archéozoologie, Archéobotanique, Sociétés Pratiques et Environnements (AASPE), CNRS, Muséum national d'Histoire naturelle, Paris, France
[2]Réserve Zoologique de la Haute Touche, 36290 Obterre, Muséum national d'Histoire naturelle, France
[3]Unité Bases de données sur la Biodiversité, Écologie, Environnement et Sociétés (BBEES), Muséum national d'Histoire naturelle, Paris, France
[4]Martin Luther University Halle-Wittenberg Central Natural Sciences Collections, Museum for domesticated animals
[5]CIRE, INRA, Nouzilly, France
[6]LRDE, INRA, Corté, France
[7]Natural History Museum of Geneva
[8]Laboratoire de Biométrie et Biologie Évolutive (LBBE), UMR 5558 CNRS, Université Lyon 1, Villeurbanne, France
[9]Institut de Systématique, Evolution, Biodiversité (ISYEB), Muséum national d'Histoire naturelle, CNRS, Sorbonne Université, EPHE, Université des Antilles, France
[10]UMR 7179, Département Adaptations du Vivant, Bâtiment d'Anatomie Comparée, CNRS, Muséum national d'Histoire naturelle, Paris, France

 DN, 0000-0001-9465-220X; IB, 0000-0002-9947-9378; RS, 0000-0003-4156-5414; SR, 0000-0002-8730-3113; AHe, 0000-0003-0991-4434; TC, 0000-0001-6021-5001

**Author for correspondence:**
Thomas Cucchi
e-mail: cucchi@mnhn.fr

Deciphering the plastic (non-heritable) changes induced by human control over wild animals in the archaeological record is challenging. We hypothesized that changes in locomotor behaviour in a wild ungulate due to mobility control could be quantified in the bone anatomy. To test this, we experimented with the effect of mobility reduction on the skeleton of wild boar (*Sus scrofa*), using the calcaneus shape as a possible

phenotypic marker. We first assessed differences in shape variation and covariation in captive-reared and wild-caught wild boars, taking into account differences in sex, body mass, available space for movement and muscle force. This plastic signal was then contrasted with the phenotypic changes induced by selective breeding in domestic pigs. We found that mobility reduction induces a plastic response beyond the shape variation of wild boars in their natural habitat, associated with a reduction in the range of locomotor behaviours and muscle loads. This plastic signal of captivity in the calcaneus shape differs from the main changes induced by selective breeding for larger muscle and earlier development that impacted the pigs' calcaneus shape in a much greater extent than the mobility reduction during the domestication process of their wild ancestors.

# 1. Introduction

Documenting the domestication process of animals in archaeology provides insights into a major cultural and biological transition in human history and into the temporal depth of its impact over biodiversity and species evolution [1–6]. Yet, tracking the domestication process in the archaeological record is an extremely challenging task for bioarchaeologists, as it involves intertwined cultural, ecological and evolutionary components [7,8], and is entirely dependent on the species involved, its domestication pathway and the intensity of their relationship with humans [9–11]. To help identify this complex and elusive process in archaeology, the process of domestication and the concept of domestic animals should be separated. The former involves the control of wild animals and may provoke no visible bioarchaeological modification, especially in the initial stages. The latter is restricted to animal populations showing clear biological modifications due to domestication [12].

To document the biological process of animal domestication, bioarchaeologists have relied on genetic markers of reproductive isolation and trait selection, or genetically induced morphological markers [13]. These morphological markers, which are all parts of an integrated 'domestication syndrome' already mentioned by Darwin [14], are associated with an animal's response to new selective pressures within a human environment [15,16], long before the selection of specific traits useful to humans [17] and are only visible after several generations [18,19]. They can be traced in the archaeological record through bone and tooth size reduction, the reduction of the volume of the brain cavity or the jaw's shortening [20]. These morphological markers have been considered as a universal trait package of animal domestication syndrome since the Russian fox farm experiment initiated by Dr Dmitri Belyaev, which demonstrated the link between the selection for tameness and the domestication syndromes. However, recent historical review called into question the ubiquity and indeed the utility of the domestication syndrome trait package [21]. Furthermore, it remains questionable for tracking early domestication processes in the archaeological record [10,22], especially when gene flow between wild and domestic populations was common [23–26], thus delaying the expression of these syndromes. So how early can bioarchaeologists document the domestication process? Can they access the plastic (non-heritable) changes immediately induced by human control over wild animal behaviour over the course of its lifetime? If such signatures of human control could be deciphered from the archaeological record, an entire new range of incipient domestication and human–animal interactions could be accessed, several millennia before the earliest visible genetically induced morphological change.

Independent of any genetic change, modifications in the use of a skeletal trait may trigger adjustments of the musculature, which can subsequently induce morphological variation through skeletal plasticity: muscle stimulation influences bone growth, development and remodelling [27,28]. To date, the 'environmental' or 'plastic' morphological responses to conditions experienced by animals under human control have received scant attention [29] despite their potential to generate fast responses in the context of domestication. Such processes could have potentially triggered a phenotypic response as early as the first generation of wild-caught animals bred or maintained in captivity. In this paper, we explore if bone plasticity can trace human-induced changes in locomotor behaviour generated by imposed restrictions on an animal's movements through captivity. The effect of several generations of captivity on the morphology of the skeleton of wild mammals (mainly felines and primates) has already been described [30–32], but many of these studies relate to the process of selection in captive populations similar to the domestication syndrome [30,33–35]. The plastic responses in the skeleton over the lifespan of a wild-caught mammal induced by captivity have been explored only in the house mouse [36]. Therefore, whether change in locomotor behaviour of a wild mammal taken out of its ecological context can have a significant impact on its bone

anatomy and how much this plastic signal can be separated from the selection on behaviour remains unknown.

To test this hypothesis, we used an experimental approach on the wild boar (*Sus scrofa*), since ungulates played a major role as food in the Neolithic transition and because few studies have used ungulates to explore the effects of captivity [30]. To control for genetic and environmental factors of skeleton variation, we captured weaned piglets from a genetically homogeneous wild boar population, few kilometres away from the experimental farm where they were reared according to two regimes of mobility reduction. The initial objectives were to explore whether captive locomotor behaviour could lead to bone shape modifications beyond the reaction norm observed in wild-caught wild boar and to quantify how much these plastic modifications are impacted by the functional link between bone and muscles. The second objective was to compare this experimental plastic response to captivity with the phenotypic changes induced by selective breeding (artificial selection) in pigs over the last 200 years. In order to compare current and past phenotypic variation and to track these markers of plasticity in the archaeological record, we chose the ankle bone (the calcaneus) as a phenotypic marker. This bone is key in terrestrial mammal locomotor behaviour, acting as a lever arm for the ankle extensors, and is subjected to high tensile, bending and compressive forces [37,38]. Furthermore, even though its proximal epiphyses fuse rather late, this bone is well preserved in archaeological contexts thanks to its compacity. Finally, it was not purposely broken to access the bone marrow, as is the case for long bones and is thus often retrieved intact [39].

# 2. Material and methods

## 2.1. Experimental design

To experimentally test the plastic response of mobility reduction on the shape of the calcaneus in a wild ungulate, we rely on a control population of wild boar living in a $100\,000\,\mathrm{m}^2$ (10 ha) fenced forest in Urciers (Indre, France), where human interaction is intentionally kept to a minimum in order to ensure that the boars' behaviour remains as natural as possible. Consequently, the mobility of this control population is not very different from the pre-Neolithic situation in the Near East where wild boar were probably partly commensal before being fully domesticated [40,41]. From this genetically homogeneous control population, we captured 24 6-month-old piglets that we divided into two groups of equal sample size and sex ratio. Both groups were raised in a zoological reserve 100 km away from the control population until the age of 24 months, in two different contexts of mobility: a $3000\,\mathrm{m}^2$ (0.3 ha) wooded pen and an indoor stall of $100\,\mathrm{m}^2$, where males and females were separated. To better control the effect of diet on the growth of the two groups, they were both supplied with standardized food pellets in order to maintain a healthy weight, according to the standard nutritional requirements of European wild boar populations [42]. Water was available ad libitum.

This experiment received full ethical agreement (APAFIS#5353-201605111133847).

## 2.2. Comparative collection

The captive signal at adulthood in wild boars reared in stalls and pens was contrasted with 28 adult wild-caught wild boars from the control population in Urciers, three other populations in France and two in Switzerland (table 1). All these specimens were wild caught between 1 and 18 years of age. We also included two captive wild boars from the museum for domesticated animals in Halle (Museum für Haustierkunde 'Julius Kühn', MHK). These specimens were captured around 1900 at an unknown age from a population of German wild boars, but we know their mobility was significantly reduced for between 10 and 15 years before their death.

To compare the plastic response of captivity with the phenotypic change driven by the last 200 years of artificial selection, we collected 19 domestic pigs, including 11 specimens from traditional landraces, which were part of a conservation programme, and eight from an intense breeding programme dedicated to industrial meat production (table 1). All these domestic specimens are part of the historical collections of the MHK. They were reared in stalls and were aged between 1 and 9 years. We also included five free-range Corsican landrace pigs (*U nustrale*) aged between 14 and 18 months. These pigs were bred according to the traditional extensive herding practice in Corsica where pigs can roam freely in large areas of maquis forest to access natural resources for their diet [43].

**Table 1.** Sample size and origin of the samples. MNHN: Muséum national d'Histoire naturelle in Paris, MHK: Museum für Haustierkunde Julius Kühn in Halle, MHNG: Muséum d'Histoire Naturelle in Geneva. For information regarding body mass, age, sex, muscles and status of the individuals included, please see electronic supplementary material, SI 1.

| status | category | population/Breed | mobility | curation | N GMM | N muscle data | N body mass | grouping factor |
|---|---|---|---|---|---|---|---|---|
| wild boar | control (France) | Urciers | wild caught | MNHN | 5 | 3 | 5 | WB_ctrl |
| wild boar | experiment (France) | Urciers | captive reared (stall) | MNHN | 12 | 10 | 12 | WB_stall |
| wild boar | experiment (France) | Urciers | captive reared (pen) | MNHN | 12 | 12 | 12 | WB_pen |
| wild boar | Switzerland | Bois de la Batie | wild caught | MHNG | 5 | 0 | 5 | WB_wc |
| wild boar | Switzerland | Dardagny | wild caught | MHNG | 2 | 0 | 2 | WB_wc |
| wild boar | France | Saint-Jean-D'Aulps | wild caught | MHNG | 6 | 0 | 6 | WB_wc |
| wild boar | France | Compiègne | wild caught | MNHN | 4 | 0 | 4 | WB_wc |
| wild boar | France | Chambord | wild caught | MNHN | 6 | 1 | 6 | WB_wc |
| wild boar | Germany | Germany | captive reared (stall) | MHK | 2 | 0 | 2 | WB_MHK |
| pigs | landraces | Bayerisches Landschwein | captive reared (stall) | MHK | 5 | 0 | 0 | DP_Land |
| pigs | landraces | Hannover-Braunschweig Landschwein | captive reared (stall) | MHK | 3 | 0 | 0 | DP_Land |
| pigs | landraces | Lincolnshire | captive reared (stall) | MHK | 1 | 0 | 0 | DP_Land |
| pigs | landraces | Tamworth | captive reared (stall) | MHK | 2 | 0 | 0 | DP_Land |
| pigs | landraces | Corsican breed | free range | MNHN | 5 | 0 | 0 | DP_Cor |
| pigs | improved breeds | Berkshire | captive reared (stall) | MHK | 4 | 0 | 0 | DP_Improv |
| pigs | improved breeds | Deutsches edelschwein | captive reared (stall) | MHK | 2 | 0 | 0 | DP_Improv |
| pigs | improved breeds | Veredeltes Landschwein | captive reared (stall) | MHK | 2 | 0 | 0 | DP_Improv |

## 2.3. Calcaneus three-dimensional models

To compare the specimens from the different institutions, we combined three-dimensional images acquired from medical CT scan and photogrammetry. Previous studies have shown that the results from these methods are comparable [44,45]. Seventy specimens were CT scanned on a Siemens Somatom® medical CT scanner with a spatial resolution of 100 to 500 micrometres at the CIRE imaging service of the INRA in Nouzilly. Sixteen specimens from Germany were scanned using the medical CT scanner of the Halle hospital with the same parameters. For each calcaneus, three-dimensional surfaces were obtained from the DICOM images stacks using Avizo v. 8.0.

The three-dimensional models of 13 specimens from Switzerland and MHK were obtained using photogrammetry. We used a Sony DSLR-A350 camera with a 50 mm lens. Each calcaneus was placed on a rigid, planar cardboard sheet with a calibrated reference pattern. The photographs were shot at regular intervals of approximately 22° near the proximal and distal extremities and 45° near the lateral and medial sides as the operator rotated the cardboard sheet. Sets of 12 pictures were acquired from three different vertical angles (approximately 10°, 40° and 70°) for both dorsal and plantar sides. Some additional pictures were shot to ensure every detail was captured. In total, more than 72 ($12 \times 3 \times 2$) pictures were used to reconstruct each model. Sets of images were processed with Agisoft PhotoScan to obtain three-dimensional surface files of each calcaneus and scaled using the 'Transform: Scale' tool in MeshLab.

## 2.4. three-dimensional shape and centroid size variables

To capture the complexity of the calcaneus form, encompassing the articulation and muscle insertion areas, we used a three-dimensional sliding semi-landmark procedure [46] (figure 1 and table 2). This included 14 anatomical type III landmarks on the maxima of curvature due to the lack of structures for placing type I or type II landmarks. A total of 181 sliding semi-landmarks on seven curves constrained by anatomical landmarks and corresponding to joint surfaces, muscle attachment surfaces and the junction with the epiphysis were also taken. Finally, 763 surface sliding semi-landmarks uniformly distributed over the surface were taken. Anatomical landmarks and sliding semi-landmarks were obtained from the three-dimensional polygonal surfaces using IDAV Landmark v. 3.0 [47]. Semi-landmarks were slid while minimizing the bending energy using the R package 'Morpho' [48]. We used a generalized Procrustes analysis [49] to rotate, translate and scale the landmark configurations to obtain a new set of shape variables (Procrustes coordinates) and the centroid size (CS) of the calcaneus using 'Morpho'. CS is the square root of the sum of squared distances of the landmarks from their centroid.

## 2.5. Life-history dataset

To analyse the covariation between life-history traits and the calcaneus form in wild boar, we collected body mass, sex and age for the wild boar specimens (electronic supplementary material, SI 1). Captive-raised wild boars have a known age at death, but wild-caught wild boars had to be aged according to their dental eruption and occlusal attrition stages [50,51].

## 2.6. Muscle force estimates

To measure the covariation between calcaneus shape and the functional properties of the muscles attached to the calcaneus, we dissected the lateral gastrocnemius (LG), the medial gastrocnemius (MG) and the soleus (S) of 22 experimental captive-reared wild boars, four specimens from the control population, and eight wild boars from French populations (table 1, electronic supplementary material, SI 1). Muscle data were not available for other specimens. The muscles were weighed to the nearest gram and muscle fascicle length was measured with calipers. Based on the known density of mammalian muscle (1.06 g cm$^{-3}$) [52], we calculated the physiological cross-sectional area (PCSA) as a proxy for muscle force by dividing the muscle volume by the fibre length [53].

## 2.7. Available area for locomotion

In order to measure the influence of mobility restriction on calcaneus shape variation in wild boars, we collected information on the available area for each of the experimental groups and the wild populations

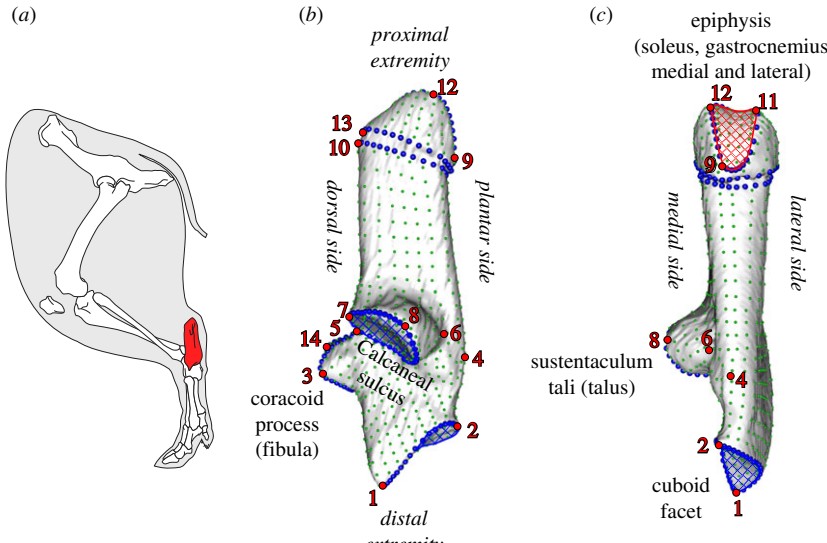

**Figure 1.** (*a*) The wild boar calcaneus in lateral view in relation to the other bones and muscles of the leg. (*b*) and (*c*) Medial and plantar views with landmarks (red dots), semi-landmarks on curves (blue dots) and semi-landmarks on surfaces (green dots). Blue shaded areas represent articular surfaces and the red shaded area represents the muscle attachment surface.

**Table 2.** Anatomical definition of the 14 landmarks (LM) and 7 curves (C).

| | |
|---|---|
| LM1 | distal end of the cuboid facet |
| LM2 | proximo-plantar end of the cuboid facet |
| LM3 | end of the beak of the coracoid process |
| LM4 | maximum of curvature of the plantar bulge on the plantar margin |
| LM5 | dorso-proximal end of the calcaneal sulcus |
| LM6 | planto-lateral end of sustentaculum tali |
| LM7 | dorsal end of the sustentaculum tali |
| LM8 | medial end of sustentaculum tali |
| LM9 | plantar end of the epiphysis |
| LM10 | dorso-proximal end of the bulge of the proximal part (not on the epiphysis) |
| LM11 | proximal end of the lateral lobe of the epiphysis (secondary lobe) |
| LM12 | proximal end of the medial lobe of the epiphysis (main lobe) |
| LM13 | dorsal end of the epiphysis |
| LM14 | dorso-proximal end of the lateral part of the coracoid process |
| C1 | edge of the articular surface of the cuboid facet |
| C2 | medial edge of the coracoid process |
| C3 | edge of the articular surface of the sustentaculum tali |
| C4 | lateral edge of the coracoid process |
| C5 | edge of the attachment surface of the tendon on the epiphysis |
| C6 | distal delineation of the junction zone between the epiphysis and the rest of the calcaneus |
| C7 | proximal delineation of the junction zone between the epiphysis and the rest of the calcaneus |

(electronic supplementary material, SI 1). The areas available for the experimental contexts (pens and stalls) were known; however, the size of enclosures for the MHK wild boars had to be estimated from archive photos. The available areas for free wild boar populations from France and Switzerland were measured using Google maps, based on the measured area of the forest in which the specimens were caught.

## 2.8. Statistical analyses

### 2.8.1. Size, shape and life traits covariation in captive and wild-caught wild boars

The difference in calcaneus size and body mass variation among captive-reared and wild-caught wild boars, taking into account their sex, was visualized with a box plot and tested with factorial analysis of variance (ANOVA) and pairwise test. The correlation between calcaneus size and body mass was tested using the Pearson test.

We tested whether the calcaneus shape differed among wild-caught and captive-reared wild boars using their status (GP1) as grouping factor while accounting for shape covarying with age, body mass and bone centroid size (allometry) using a factorial MANCOVA with 1000 permutations.

We visualized and tested the PCSA differences among wild-caught and captive-reared wild boars using a box plot and ANOVA. The correlations between PCSA and calcaneus centroid size and body mass were tested with a regression and the Pearson test. For the visualization of shape deformation associated with body mass, calcaneus centroid size, available area for locomotion (in $m^2$) and intrinsic muscle force (PCSA), we used partial least squares (PLS) analyses [54,55]. For the available area of mobility, as every specimen from each group had the same available area, we used group means for the PLS analysis.

### 2.8.2. Comparing plastic size and shape response to captivity with changes induced by artificial selection

Size differences between wild boars and domestic pig groups were tested using an ANOVA with the pairwise comparison tests (Bonferroni correction) and visualized with a box plot. The shape differences were tested with a Procrustes ANOVA and visualized using a Canonical Variate Analysis (CVA); both were performed on a shape-reduced dataset after a principal component analysis (PCA) performed on the Procrustes coordinates to keep 95% of the variance [56]. To visualize shape deformations along the canonical axes, we calculated the theoretical minimum and maximum shapes for each axis, associated with a heatmap on landmarks corresponding to the distance between the minimum and maximum of the axis. We also visualized the deformation along two vectors using the mean shape of wild-caught wild boars as a reference, towards the mean shape of captive-reared wild boars as the first vector and the mean shape of domestic pigs as the second vector.

To test the shape difference depending on the main categories (GP2: wild boars/traditional pig breeds/improved pig breeds) and their sex, including the interaction of these two factors, we used a factorial MANOVA with a 1000 permutations procedure.

All the statistics were performed using R [57]. Factorial MANOVA, Procrustes ANOVA and PLS were performed using the R package 'geomorph' [58]. ANOVA, PCA were performed using the package 'stats' [57], CVA using the package 'Morpho' [48] and visualizations were performed by using 'Morpho', 'stat' and 'plotrix' [59].

## 3. Results

### 3.1. Plastic responses to captivity in wild boars

We found no significant differences in body mass (ANOVA: $F = 1.171$, $p = 0.33$) and calcanaeum centroid size (ANOVA: $F = 0.673$, $p = 0.57$) between the wild-caught and captive-reared wild boars. A slight trend towards greater bone size and body mass in captive-reared specimens compared to wild boars can be observed. Yet, no difference exists among the two captive-reared groups (figure 2). Sexual dimorphism is significant in calcaneus size (ANOVA: $F = 14.385$, $p = 0.0004$) but not in body mass (ANOVA: $F = 3.2$, $p = 0.08$). This absence of sexual dimorphism in body mass was mainly true for captive-reared specimens in stalls (figure 2b). The correlation between calcaneus centroid size and body mass was overall very strong (Pearson correlation coefficient: 0.82, $p < 0.0001$). Yet, stall-reared specimens display weaker correlation than wild-caught and pen-reared specimens (figure 2c).

Shape differences were significant between wild-caught and captive-reared wild boars ($F = 2.1902$, $p = 0.015$) as well as age effect over shape ($F = 2.1597$, $p = 0.012$), body mass ($F = 2.8195$, $p = 0.003$) and CS ($F = 3.5100$, $p = 0.001$). The three groups of wild-caught and captive-reared wild boars show a common shape covariation with age (GP1 × age: $F = 0.0332$, $p = 0.476$) and centroid size (GP1 × size: $F = 1.0442$, $p = 0.163$), but differed in the way body mass impact the calcaneus shape (GP1 × mass: $F = 2.7302$, $p = 0.001$).

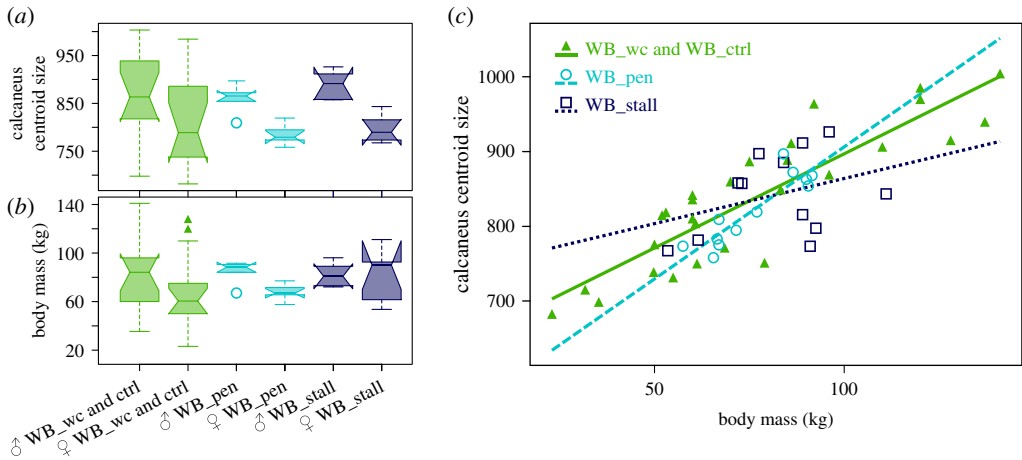

**Figure 2.** Differences among captive-reared and wild-caught wild boars in (*a*) calcaneus centroid size and (*b*) body mass. (*c*) Regression between calcaneus centroid size and body mass among wild-caught and captive-reared wild boars.

Compared to the wild-caught wild boars, the captive-reared wild boars had significantly greater muscle PCSA for the lateral (figure 3*a*, ANOVA: mean square = 19.732, $F = 8.63$, $p = 0$ 0.0015) and medial (figure 3*b*, ANOVA: mean square = 18.716, $F = 12.754$, $p = 0$ 0.0001) gastrocnemius, but not for the soleus (figure 3*c*, ANOVA: mean square = 2.807, $F = 2.866$, $p = 0.075$). Sexual dimorphism in muscle PCSA among wild-caught and captive-reared wild boars was significant only for the lateral gastrocnemius (figure 3*c*, ANOVA: $F = 6.518$, $p = 0.0169$).

Similar calcaneus shape changes were found in association with body mass and calcaneus centroid size increase (figure 4*a,c*). Changes are localized in the distal part of the calcaneus with a more dorso-plantarily curved calcaneus and a shift of sustentaculum tali towards the distal extremity. The mobility reduction (figure 4*b*) and a greater force generation potential of the lateral and medial gastrocnemius muscles (figure 4*d*) is associated with a more elongated epiphysis that is curved towards the dorsal side and a more distally shifted sustentaculum tali.

## 3.2. Comparing plastic response to captivity in wild boars to artificial selection signal in domestic pigs

The calcaneus CS (figure 5) differed significantly among the groups (ANOVA: $F = 5.75$, $p < 0.001$). All the domestic pigs from the traditional and improved breeds had a larger calcaneus than wild-caught and captive-reared wild boars, except for the Corsican free-ranging pigs which were intermediate and showed no significant differences with the landraces, improved breeds or the wild-caught and captive-reared wild boars. The German captive-reared wild boars were in the upper range of wild-caught wild boars, probably because they are old specimens and their different genetic pool. These results suggest that calcaneus size increase is not affected by a change in locomotor behaviour but rather by artificial selection.

The calcaneus shape differed significantly among the eight groups of wild boars and domestic pigs (MANOVA: $F = 3.0564$; $p < 0.0001$). The main shape differentiation shown on the first canonical axis (figure 6*a*) was driven by the divergence between wild boars and pig breeds, with modern breeds being the most divergent in shape. Factorial MANOVA showed that difference between wild boars/ traditional pig breeds/modern breeds accounted for 19% of the total variance of calcaneus shape in adults ($p < 0.01$), while sexual dimorphism had almost no effect on shape variation and was similar in both wild boars and domestic breeds (electronic supplementary material, SI 2). The shape deformation associated with the divergence between wild-caught wild boars and domestic pigs (figure 6*b,d*) is marked by an increase in calcaneus robustness and thickness with a medio-laterally wider calcaneus, particularly on the medial side at the junction with the epiphysis where we can see a bulge. The coracoid process and the cuboid facet are also more robust and the epiphysis is longer in domestic pigs than in wild boars.

The second canonical axis is related to the difference in mobility, expressed both in wild boars and pigs (figure 6*a*). The captive-reared wild boars were strongly divergent from their wild-caught

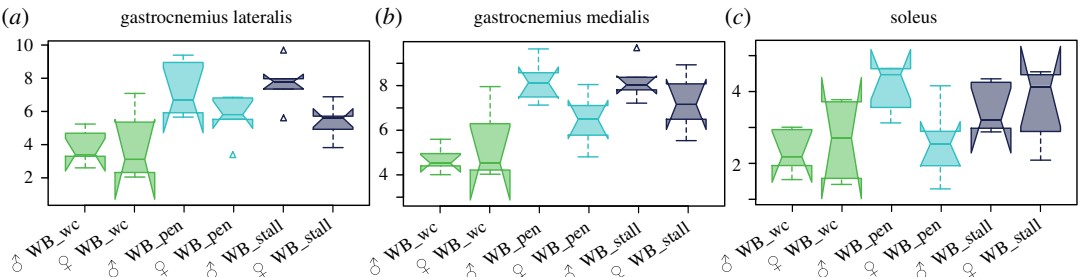

**Figure 3.** Difference in muscle PCSA among wild-caught and captive-reared wild boars taking into account their sex for (*a*) the lateral gastrocnemius, (*b*) the medial gastrocnemius and (*c*) the soleus.

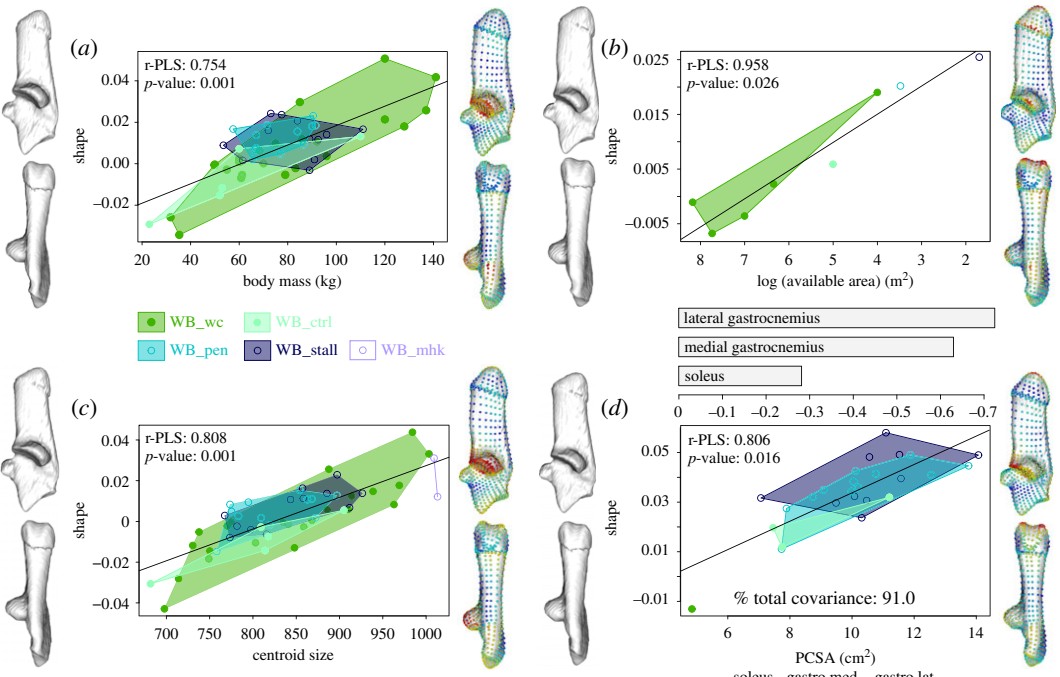

**Figure 4.** Relationships between the calcaneus shape and four continuous variables: (*a*) body mass (kg), (*b*) log (available area for mobility) (m²), (*c*) calcaneus centroid size and (*d*) muscle PCSA (cm²). Wild-caught wild boars are visualized in green and captive-reared wild boars in blue. Black lines represent the PLS regression line. Shape deformations are visualized with negative extreme shapes on the left of the scatterplot and positive extreme shapes are on the right. The heatmap on landmarks for positive extreme shapes represents the distance of each landmark from the negative extreme shape—the greater the distance the hotter the colour.

relatives along this axis, whereas the difference between the stall- and pen-reared wild boars was not significant (MANOVA, $F = 5.1959$; $p = 0.1736$). In pigs, the free-roaming Corsican pigs are the most divergent from the captive, modern breeds. The shape divergence on CV2 (figure 6*c*) and along the vector between wild-caught and captive-reared boars mean shapes (figure 6*d*) was localized on the proximal epiphysis, which was longer and curved towards the dorsal side in captive specimens. The sustentaculum tali was also slightly shifted towards the distal end compared to wild-caught wild boars.

# 4. Discussion

## 4.1. Plastic response to captivity

Previous studies have shown that several generations of captivity-induced changes in diet, exercise and stress could impact the morphology and physiology of captive populations [30]. However, most of these studies have investigated the phenotypic effect as an equivalent of the domestication syndrome, combining relaxed natural selection and unconscious selection for tameness [30,34,60]. Our

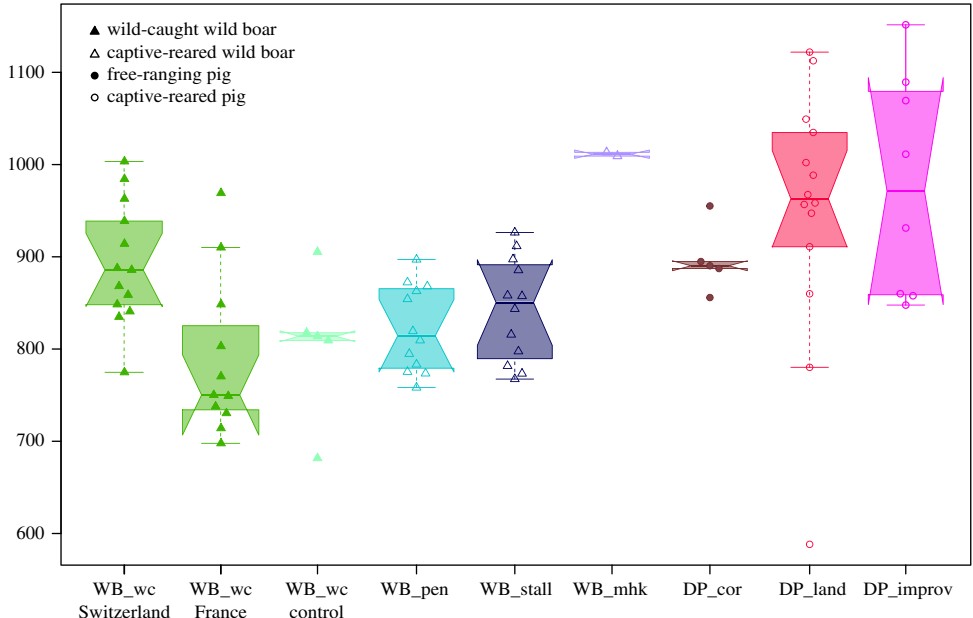

**Figure 5.** Box plot representation of the calcaneus centroid size variation in wild boars and pigs from free-ranging or captive environments.

experimental results provide the first evidence that the growth of a wild ungulate in a captive environment, where its locomotor behaviour is altered, impacts the shape of its calcaneus beyond the reaction norm of wild-caught populations. This evidence shows that morphological change in captivity can be driven by phenotypic plasticity and is probably a reflection of changes in the biomechanical environment of an animal during the course of its lifetime. These results contradict the only other study on the impact of captivity on the scapula of chimpanzees, which found no scapula shape differences among wild and captive extant populations of great apes [32], but they are in agreement with the study evidencing no foot size reduction between wild-caught and captive-reared house mice (*Mus musculus*) [36].

Indeed, we found no bone size reduction in captive-reared wild boars, suggesting that ankle bone shape has greater plasticity than overall bone size when a wild animal is facing modifications of its locomotor environment during its growth. Skeleton size reduction has been evidenced in captive black-footed ferrets (*Mustela nigripes*) after 10 years of captivity with increased mobility reduction over time [33]. This suggests that skeleton size reduction would only occur after several generations of selection for phenotypes that maximize fitness in captivity. Plastic bone size reduction in captivity is probably more a consequence of food shortages and overcrowding [61,62]. On the other hand, a plastic size increase is more likely to happen in captivity due to unrestricted access to food, driving faster growth as shown in captive chimpanzees [63].

## 4.2. Captivity and change in muscular function

The plastic imprint of captivity on the wild boar calcaneus, associated with changes of mobility reduction and an increase in intrinsic muscle force, clearly suggests that it is driven by an increase in the force exerted by the muscles on the bone when mobility reduction prevents the expression of full locomotor behaviour. The association between reduced mobility and increased muscle forces could seem counterintuitive at first. Indeed, in the wild, wild boar locomotor behaviour includes different types of movements (feeding, fleeing and dispersal) [64,65], which involve running fast up to 40 km h$^{-1}$, jumping high up to 150 cm [66] and daily travel, typically under 10 km but which can reach up to 80 km in one night [67]. Compared to the locomotor behaviour in captivity, where the need to search for food, flee predators, disperse or compete is suppressed, one would expect that the greater range of movements of free-ranging wild boars would have produced stronger gastrocnemius and soleus muscles. The answer possibly lies in the stereotyped locomotor behaviour induced by captivity, which may involve a reorientation of the calcaneus position due to more upright standing and thus more activity in the ankle extensors inducing greater stress on the

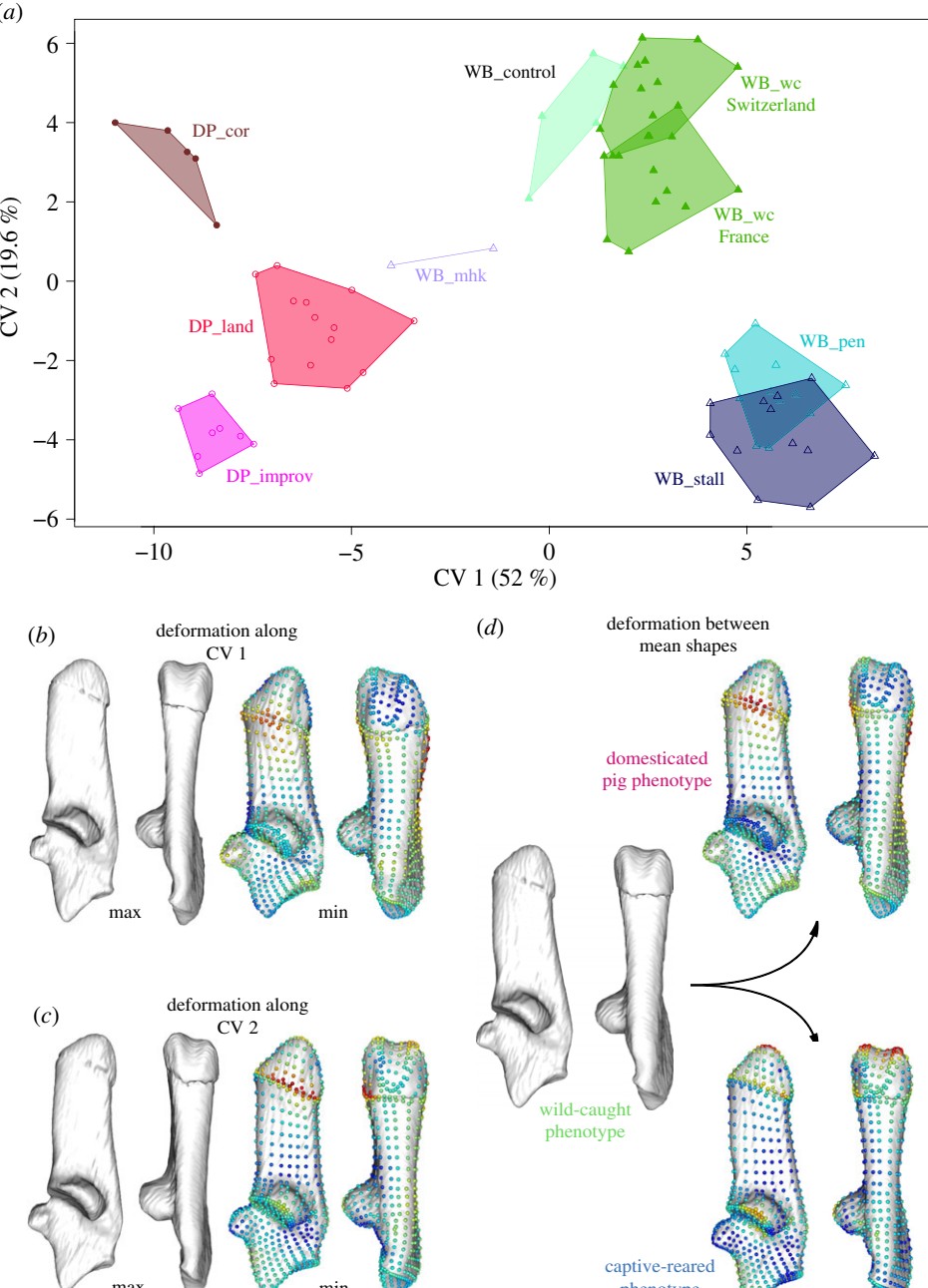

**Figure 6.** Morphospace based on a CVA representing the pattern of shape divergence among adult wild boars and pigs living in free-ranging or captive environments (*a*). Shape deformations correspond to the difference between the theoretical maximum and minimum shapes for axis 1 (*b*) and 2 (*c*). For better visibility, the extreme shapes were amplified by a factor of two and visualized from the maximum to the minimum. Calcaneus shapes are represented in medial and plantar views. The heatmap superimposed on landmarks for the minimum shape represents the distance of each landmark from the maximal shape: blue indicates a shorter distance and red indicates a greater distance. Shape deformations are also visualized between three mean shapes (*d*): wild-caught, captive-reared and pigs.

calcaneus [68]. Reorientation of the calcaneus shape has been observed in cervids living in closed environments that used fast-twitch muscles for more saltatorial locomotion compared to those living in open environments which use slow-twitch muscles for more cursorial locomotion [69]. Additionally, changes in the type of movement from long distance to slower movements requiring slower oxidative muscles—characterized by a lower cross-sectional fibre area versus more burst-type short distance locomotion in captivity associated with fast-contracting muscles involving a greater muscle fibre cross-sectional area—may drive this pattern.

## 4.3. Plastic versus selective changes

These plastic shape differences in captive-reared wild boars follow a different phenotypic trajectory compared to the shape divergence between wild-caught and domestic phenotypes induced by the last 200 years of selective pig breeding. However, this signal is also expressed in pigs, since we found a divergence between captive-reared and free-ranging pig breeds, suggesting that this plastic signal is of a more general nature. The patterns of covariation we observed between calcaneus shape and body mass, bone centroid size, available space for movements and muscle force suggest that shape divergence between domestic breeds and wild boar phenotypes may have been driven by increases in overall body mass and associated bone size (figure 4). Selection in breeding programmes for larger and heavier individuals [70–72] through growth rate, carcass length and muscle production [68] has produced a very different set of external loads on the bones [71], probably mediated by a different locomotion posture [73] compared to wild boars. Both the increased loads from the increased body mass and the muscle forces generated during locomotion inducing greater bone mass and strength [74] could explain the change in calcaneus shape towards greater shape robusticity observed in pigs.

# 5. Conclusion

Our results provide strong evidence that changes in locomotor behaviour induced by captivity can impact the bone anatomy of a wild ungulate over the course of its lifetime. This plastic imprint of captivity in wild boar calcaneus produces a phenotype beyond the usual reaction norm in natural environments but along a different trajectory than the divergence driven by selective breeding. If we consider that human control and modification of a wild animal's movements is one of the first steps towards domestication, our results provide new methodological perspectives for bioarchaeological approaches linking plastic responses to the domestication process involving cultural control of wild animal movements. Further studies now need to explore when these changes are implemented during the growth to understand the developmental processes behind these plastic changes.

Data accessibility. Code, data and metadata to perform statistical analyses on the calcaneus form and muscle PCSA are archived in the Dryad Digital Repository (http://datadryad.org/) and are currently accessible at the following location: https://doi.org/10.5061/dryad.1zcrjdfnr.

Authors' contributions. T.C. designed the research with A.He. and J.-D.V.; Y.L., K.O., B.B. and T.C. conducted the experimental fieldwork; T.C. collected the CT data with F.Le.; H.H. collected the photogrammetry data and A.He. collected the muscular data; F.La., I.B., C.C. and T.C. created the database of the project; H.H. carried out the GMM analyses and interpreted the data with T.C., A.He., R.C., D.N. and S.R.; T.C. and H.H. led the manuscript with scientific and editorial input from A.He., S.R., J.-D.V., D.N., R.S., A.Ha., F.C. and J.S. All authors gave final approval for publication.

Competing interests. We declare we have no competing interests.

Funding. This research has been funded by the ANR, through the Domexp project (ANR-13-JSH3-0003-01), and the LabEx ANR-10-LABX-0003-BCDiv, in the programme 'Investissements d'avenir' ANR-11-IDEX-0004-02. This project has also benefited from financial supports of the Muséum national d'Histoire naturelle (Paris) and the CNRS INEE (Institut écologie et environnement).

Acknowledgements. We are most grateful to the director (Roland Simon) of the Réserve Zoologique de la Haute-Touche and its staff (Christophe Audureau, Jérémy Bernard, Jérémy Coignet, Christophe Jubert, Fabien Kurek, Sandrine Laloux, Emmanuel Marechal, Régis Rabier, Patrick Roux, Colin Vion) for their help during the set-up of the experimental structures and during the data acquisition, and the care they provided to the experimental specimens. Without their dedication and expertise, this research would not have been possible. We thank the CIRE platform at the Institut National de la Recherche Agronomique (INRA) in Nouzilly for the CT data acquisition (François Lecompte, Hans Adriensen) and their technical team (Frédéric Elbout, Christian Moussu and Luc Perrigouard) for their support during the CT acquisition and the handling of animals. We would like to thank Jill Cucchi for the copy editing of the manuscript.

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
