## [Reviewer comments · Royal Society Open Science]

Review History

RSOS-192039.R0 (Original submission)

Review form: Reviewer 1

Is the manuscript scientifically sound in its present form?

Yes

Are the interpretations and conclusions justified by the results?

Yes

Is the language acceptable?

Yes

Do you have any ethical concerns with this paper?

No

Have you any concerns about statistical analyses in this paper?

No

Recommendation?

Accept with minor revision (please list in comments)

Comments to the Author(s)

This paper represents an important contribution to the study of domestication as an evolutionary process. It provides a powerful new methodological tool that has the potential to re-orient our understanding of the process of domestication from the biological results, to the behavioral impetus. The authors succeed in providing credible evidence for plastic responses in the skeletal morphology of *S. scrofa* resulting from human-driven changes in locomotor behaviour. By focusing on and providing evidence for the effects of confinement on wild populations--likely one of the earliest human behaviors driving the evolutionary transition we recognize as domestication--this paper not only provides a specific and powerful new tool for identifying one form of early human intervention setting the stage for evolutionary change, but it also provides a model for archaeologists and other researchers interested in domestication for studies that more closely account for how context and contingency structure both the process itself and its outcome, which has historically received insufficient attention. I commend the authors for the scientifically sound and forward-thinking research design they present here, and am convinced and thrilled by their results. Therefore, it is with great enthusiasm that I recommend this paper for publication.

I have only one, minor and rather semantic comment, that I think should be considered by the authors when revising the manuscript. This concerns only the theoretical framing of the experiment. In the introduction, second paragraph I urge the authors to revisit their use of "domestication syndrome" (lines 52-58). Recent re-evaluation of the concept by Lord et al. (in press, CellPress Reviews) has called into question the ubiquity and indeed the utility of the domestication syndrome trait-package. While the nature of the experiment makes it clear that the authors of this manuscript in review are not limiting their study of domestication to the narrow genetic confines imposed by reliance on this trait package, by stating they are "all shared among domestic species, regardless of their phylogenetic relationships" they seem to reify it. The problem with the domestication syndrome is less that the traits associated with it are "questionable for tracking early domestication"--although this is certainly true as well--but more that such traits have been long assumed as universal markers at all. As phrased, the use of the domestication syndrome as a means to highlight the dearth of attention that plastic responses and selective contexts have received comes off as a bit of a straw man. A more nuanced treatment that at minimum gives a nod to the controversy and debate surrounding the validity of such a syndrome is needed here.

Review form: Reviewer 2

Is the manuscript scientifically sound in its present form?

Yes

Are the interpretations and conclusions justified by the results?

Yes

Is the language acceptable?

Yes

Do you have any ethical concerns with this paper?

No

Have you any concerns about statistical analyses in this paper?

No

Recommendation?

Accept with minor revision (please list in comments)

Comments to the Author(s)

I have a few observations which the authors could consider addressing in the paper if appropriate.

1) One of their wild caught boar samples appears to have come from the same reserve (Urciers) as their experimental animals. This is very important in validating the actual experiment (i.e. the only wild-caught control for environment, altitude,vegetaion etc) but not really highlighted enough in the text.

2) Following on from 1), the wild caught boar from other parts of France and Switzerland were likely from different environments etc, but all the data is pooled (understandably) when comparing with experimental and domestic data. I know the sample sizes are small, but were there any differences in shape between these different wild boar populations?

3) It's interesting that the Corsican free-range pigs are at the extreme end of shape divergence (Fig 6a). Not much further is said about this. Not what I would have expected, so what might be the explanation?

4) I'm intrigued by the counter-intuitive (but well-explained in the paper) discussion about reduced mobility and additional increased muscle forces, along with no evidence for size decrease in captivity - but likely an initial increase). Does this perhaps support/explain evidence presented by others using GMM approaches to explore pig domestication, who have claimed the presence of a 'domestic' tooth shape in early Neolithic suids associated with large sized animals indistinguishable in size to wild boar?

Decision letter (RSOS-192039.R0)

27-Jan-2020

Dear Dr Cucchi

On behalf of the Editors, I am pleased to inform you that your Manuscript RSOS-192039 entitled "The mark of captivity: plastic responses in the ankle bone of a wild ungulate (*Sus scrofa*)" has been accepted for publication in Royal Society Open Science subject to minor revision in accordance with the referee suggestions. Please find the referees' comments at the end of this email.

The reviewers and handling editors have recommended publication, but also suggest some minor revisions to your manuscript. Therefore, I invite you to respond to the comments and revise your manuscript.

- Ethics statement

- Data accessibility

It is a condition of publication that all supporting data are made available either as supplementary information or preferably in a suitable permanent repository. The data accessibility section should state where the article's supporting data can be accessed. This section should also include details, where possible of where to access other relevant research materials such as statistical tools, protocols, software etc can be accessed. If the data has been deposited in

an external repository this section should list the database, accession number and link to the DOI for all data from the article that has been made publicly available. Data sets that have been deposited in an external repository and have a DOI should also be appropriately cited in the manuscript and included in the reference list.

<http://datadryad.org/submit?journalID=RSOS&manu=RSOS-192039>

- **Competing interests**

- **Authors' contributions**

- **Acknowledgements**

- **Funding statement**

Because the schedule for publication is very tight, it is a condition of publication that you submit the revised version of your manuscript before 05-Feb-2020. Please note that the revision deadline will expire at 00.00am on this date. If you do not think you will be able to meet this date please let me know immediately.

When submitting your revised manuscript, you will be able to respond to the comments made by the referees and upload a file "Response to Referees" in "Section 6 - File Upload". You can use this

to document any changes you make to the original manuscript. In order to expedite the processing of the revised manuscript, please be as specific as possible in your response to the referees. We strongly recommend uploading two versions of your revised manuscript:

If your manuscript is newly submitted and subsequently accepted for publication, you will be asked to pay the article processing charge, unless you request a waiver and this is approved by Royal Society Publishing. You can find out more about the charges at <https://royalsocietypublishing.org/rsos/charges>. Should you have any queries, please contact openscience@royalsociety.org.

on behalf of Professor Marcelo Sanchez (Associate Editor) and Kevin Padian (Subject Editor)
 openscience@royalsociety.org

Reviewer comments to Author:

Reviewer: 1

Comments to the Author(s)

This paper represents an important contribution to the study of domestication as an evolutionary process. It provides a powerful new methodological tool that has the potential to re-orient our understanding of the process of domestication from the biological results, to the behavioral impetus. The authors succeed in providing credible evidence for plastic responses in the skeletal morphology of *S. scrofa* resulting from human-driven changes in locomotor behaviour. By focusing on and providing evidence for the effects of confinement on wild populations--likely one of the earliest human behaviors driving the evolutionary transition we recognize as domestication--this paper not only provides a specific and powerful new tool for identifying one form of early human intervention setting the stage for evolutionary change, but it also provides a model for archaeologists and other researchers interested in domestication for studies that more closely account for how context and contingency structure both the process itself and its outcome, which has historically received insufficient attention. I commend the authors for the scientifically sound and forward-thinking research design they present here, and am convinced and thrilled by their results. Therefore, it is with great enthusiasm that I recommend this paper for publication.

I have only one, minor and rather semantic comment, that I think should be considered by the authors when revising the manuscript. This concerns only the theoretical framing of the experiment. In the introduction, second paragraph I urge the authors to revisit their use of "domestication syndrome" (lines 52-58). Recent re-evaluation of the concept by Lord et al. (in press, CellPress Reviews) has called into question the ubiquity and indeed the utility of the domestication syndrome trait-package. While the nature of the experiment makes it clear that the authors of this manuscript in review are not limiting their study of domestication to the narrow genetic confines imposed by reliance on this trait package, by stating they are "all shared among domestic species, regardless of their phylogenetic relationships" they seem to reify it. The problem with the domestication syndrome is less that the traits associated with it are "questionable for tracking early domestication"--although this is certainly true as well--but more that such traits have been long assumed as universal markers at all. As phrased, the use of the domestication syndrome as a means to highlight the dearth of attention that plastic responses and selective contexts have received comes off as a bit of a straw man. A more nuanced treatment that at minimum gives a nod to the controversy and debate surrounding the validity of such a syndrome is needed here.

Reviewer: 2

Comments to the Author(s)

I have a few observations which the authors could consider addressing in the paper if appropriate.

1) One of their wild caught boar samples appears to have come from the same reserve (Urciers) as their experimental animals. This is very important in validating the actual experiment (i.e. the only wild-caught control for environment, altitude, vegetation etc) but not really highlighted enough in the text.

2) Following on from 1), the wild caught boar from other parts of France and Switzerland were likely from different environments etc, but all the data is pooled (understandably) when comparing with experimental and domestic data. I know the sample sizes are small, but were there any differences in shape between these different wild boar populations?

3) It's interesting that the Corsican free-range pigs are at the extreme end of shape divergence (Fig 6a). Not much further is said about this. Not what I would have expected, so what might be the explanation?

4) I'm intrigued by the counter-intuitive (but well-explained in the paper) discussion about reduced mobility and additional increased muscle forces, along with no evidence for size decrease in captivity - but likely an initial increase). Does this perhaps support/explain evidence presented by others using GMM approaches to explore pig domestication, who have claimed the presence of a 'domestic' tooth shape in early Neolithic suids associated with large sized animals indistinguishable in size to wild boar?

Author's Response to Decision Letter for (RSOS-192039.R0)

See Appendix A.

Decision letter (RSOS-192039.R1)

07-Feb-2020

Dear Dr Cucchi,

It is a pleasure to accept your manuscript entitled "The mark of captivity: plastic responses in the ankle bone of a wild ungulate (*Sus scrofa*)" in its current form for publication in Royal Society Open Science. The comments of the reviewer(s) who reviewed your manuscript are included at the foot of this letter.

on behalf of Professor Marcelo Sanchez (Associate Editor) and Kevin Padian (Subject Editor)
openscience@royalsociety.org

Appendix A

Reply to reviewers of the manuscript Manuscript RSOS-192039

Title:

The mark of captivity: plastic responses in the ankle bone of a wild ungulate (*Sus scrofa*)

Authors:

Hugo Harbers¹, Dimitri Neaux¹, Katia Ortiz², Barbara Blanc², Flavie Laurens, Isabelle Baly, Renate Schafberg³, Ashleigh Haruda³, François Lecompte⁴, François Casabianca⁵, Jacqueline Studer⁶, Sabrina Renaud⁷, Raphael Cornette⁸, Yann Locatelli², Jean-Denis Vigne¹, Anthony Herrel⁹, Thomas Cucchi^{1*}.

* Corresponding author: cucchi@mnhn.fr

Dear Editor,

We are most grateful to the two reviewers that have provided very constructive and insightful comments and suggestions. We believe that we have succeeded in tackling all their queries and produced a new version where all the changes have been highlighted in yellow. We hope that this new version shall meet the standard of *Royal Society Open Science*.

We have also included three authors that have contributed to the database system of the project (Isabelle Baly, Flavie Laurens) and to the CT scan acquisition process (François Lecompte).

We will reply to each of the reviewers in the following section. Our reply will be in BOLD RED.

Reviewer 1: I have only one, minor and rather semantic comment, that I think should be considered by the authors when revising the manuscript. This concerns only the theoretical framing of the experiment. In the introduction, second paragraph I urge the authors to revisit their use of "domestication syndrome" (lines 52-58). Recent re-evaluation of the concept by Lord et al. (in press, CellPress Reviews) has called into question the ubiquity and indeed the utility of the domestication syndrome trait-package. While the nature of the experiment makes it clear that the authors of this manuscript in review are not limiting their study of domestication to the narrow genetic confines imposed by reliance on this trait package, by stating they are "all shared among domestic species, regardless of their phylogenetic relationships" they seem to reify it. The problem with the domestication syndrome is less that the traits associated with it are "questionable for tracking early domestication"--although this is certainly true as well--but more that such traits have been long assumed as universal markers at all. As phrased, the use of the domestication syndrome as a means to highlight the dearth of attention that plastic responses and selective contexts have received comes off

as a bit of a straw man. A more nuanced treatment that at minimum gives a nod to the controversy and debate surrounding the validity of such a syndrome is needed here.

Reply: We completely agree with this suggestion and are most thankful to the reviewer for raising this point and providing this reference from TREE that we didn't know. We have rephrased the introduction from l. 51 to l. 61 in order to nuance even further the relevance of the domestication syndrome package to track the domestication process in archaeology.

Reviewer: 2

1) One of their wild caught boar samples appears to have come from the same reserve (Urciers) as their experimental animals. This is very important in validating the actual experiment (i.e. the only wild-caught control for environment, altitude, vegetation etc) but not really highlighted enough in the text.

We are grateful to reviewer 2 for bringing our attention to this lack of clarity in our presentation of the experimental design. We have indeed used a population for our experiment that would control for genetic and geoclimatic environment (altitude, vegetation, precipitation, Temperature etc...). We have change the description of the experiment ll.108-121 in order to make the control design clearer.

2) Following on from 1), the wild caught boar from other parts of France and Switzerland were likely from different environments etc, but all the data is pooled (understandably) when comparing with experimental and domestic data. I know the sample sizes are small, but were there any differences in shape between these different wild boar populations?

Indeed sample size for each population is too small to analyse them separately in the framework of a discriminant analysis. In order to fulfill this request we have separated French from Switzerland wild boars populations and produced reviewed diagram for calcaneus centroid size (Figure 5) and shape (figure 6) displaying these differences. These figures show that wild boars populations from Switzerland have larger calcaneus size and also different calcaneus shape from the French populations although this differences are way below the difference between wild caught and captive wild boars populations.

3) It's interesting that the Corsican free-range pigs are at the extreme end of shape divergence (Fig 6a). Not much further is said about this. Not what I would have expected, so what might be the explanation?

We were also expecting that the corsican breed would fall between wild caught wild boars and domestic phenotype in the morphospace. Although we are fully certain that the specimens are raised in a traditional free ranging system, we cannot be fully certain that they are free from genetic introgression from improved breeds (large white). Selective breeding being the main factor of divergence in this morphospace, this could explain why they are within the divergent trajectory of the other pig breeds, despite their free ranging behavior. The later is expressed on the second axis were they diverge from the non-free ranging pigs.

4) I'm intrigued by the counter-intuitive (but well-explained in the paper) discussion about reduced mobility and additional increased muscle forces, along with no evidence for size decrease in captivity - but likely an initial increase). Does this perhaps support/explain evidence presented by others using GMM approaches to explore pig domestication, who have claimed the presence of a 'domestic' tooth shape in early Neolithic suids associated with large sized animals indistinguishable in size to wild boar?

Many thanks for this question. It is a difficult one to answer because the two signals are not fully comparable. Dental form variation is far less prone to ecophenotypic plasticity compared to bones. The specimens with a domestic tooth shape and a wild boar size could be due to feralisation. They could have kept a dental shape signature from their domestic ancestries but acquired a size variation with the return to a behavior out of human control. We have tested this hypothesis using Isotopes profiles (Carbon and Nitrogen) of “large domestic” from Chalcolithic Romania showing that despite their domestic shape, the specimens fall within the spectrum of the free ranging wild boars suggesting that they behave as wild (Balasse et al., 2019).

Reference:

**Balasse, M., Cucchi, T., Evin, A., 2019. Wild game or farm animal? Tracking human-pig relationships in ancient times through stable isotope analysis, in: Hybrid Communities. Stépanoff, C & Vigne, J.D., London.
<https://doi.org/10.4324/9781315179988-12>**